# Laboratory-confirmed respiratory viral infection triggers for acute myocardial infarction and stroke: Systematic review protocol

Tu Quan Nguyen[ID][1,2,3]*, Diana Vlasenko[2,3], Aishwarya N. Shetty[2,3], Christopher M. Reid[4,5], Hazel J. Clothier[1,2,3,6], Jim P. Buttery[1,2,3,7]

1 Department of Paediatrics, The University of Melbourne, Melbourne, Victoria, Australia, 2 Centre for Health Analytics, Melbourne Children's Campus, Melbourne, Victoria, Australia, 3 Epi-Informatics Research Group, Murdoch Children's Research Institute, Melbourne, Victoria, Australia, 4 School of Population Health, Curtin University, Perth, Western Australia, Australia, 5 Centre for Cardiovascular Research and Education in Therapeutics, Monash University, Melbourne, Victoria, Australia, 6 School of Population and Global Health, The University of Melbourne, Melbourne, Victoria, Australia, 7 Department of Infectious Diseases, Royal Children's Hospital, Melbourne, Victoria, Australia

* tu.nguyen@mcri.edu.au

**Data Availability Statement:** No datasets were generated or analysed during the current study. All relevant data from this study will be made available

## Abstract

### Background

Cardiovascular disease contributes substantially to global mortality and morbidity. Respiratory tract infections, particularly influenza, may trigger an increase in the short-term risk of acute myocardial infarction (AMI) and stroke. Recent studies have also linked this risk to other respiratory viruses, including respiratory syncytial virus (RSV) and severe acute respiratory syndrome coronavirus 2 (SARS-CoV-2). However, the pathogen-specific relative contributions, the strength of their associations, and overall public health significance are poorly understood. Assuming causal links, understanding, quantifying, and comparing the effects of different pathogens as triggering factors for acute cardiovascular events is critical to guide future research and prevention. Our aim is to conduct a systematic review to examine the relative effects of laboratory-confirmed respiratory virus infections as triggers for acute myocardial infarction and stroke.

### Methods

We will conduct a comprehensive search of Ovid MEDLINE, PubMed, Ovid Embase, Cochrane Library Central Register of Controlled Trials, and Web of Science, from inception to the end of March 2024. Studies capturing respiratory viral infection(s) using laboratory-confirmatory methods, incidence of AMI or stroke (ischaemic or haemorrhagic), and those involving human participants in any country, will be assessed for eligibility. We will include the following analytical epidemiological study types: randomised controlled trials, cohort and case-control studies, self-controlled case series, and case-crossover designs. We will not impose restrictions on the date, language, study population, geographical region, or sample

upon study completion as part of the manuscript or its supporting information, with the final published review.

**Funding:** Tu Q Nguyen was supported by an Australian Government Research Training Program (RTP) PhD Scholarship and a Murdoch Children's Research Institute Top-Up Scholarship during the preparation of this manuscript. Funding sources have no role in development of the protocol, conduct of the systematic review, writing up or decision to submit for publication.

**Competing interests:** The authors declare that they have no competing interests.

size, to minimise the risk of introducing biases. Search results will be screened for eligibility by two independent reviewers, and discrepancies resolved by consensus and/or arbitration by a third reviewer. We will assess the risk of bias among the included studies by adopting the Cochrane Collaboration tools for randomised and non-randomised studies. The overall quality of studies will be assessed using the Grading of Recommendations, Assessment, Development and Evaluations (GRADE) approach. We will examine sources of heterogeneity, and if studies are sufficiently homogeneous, a meta-analysis will be conducted to calculate the pooled effect sizes. Reporting will adhere to the Preferred Reporting Items for Systematic Reviews and Meta-Analyses (PRISMA) guidelines.

## Registration

International Prospective Register of Systematic Reviews (PROSPERO) registration number: CRD42024494997.

## Introduction

### Rationale

Cardiovascular disease (CVD), particularly acute myocardial infarction (AMI) and stroke, remains the world's leading contributor of disease burden and death [1]. Despite progress in age-standardised mortality rates, the prevalence and incidence of acute cardiovascular events is set to rise due to increasing urbanisation, population growth and ageing [2]. Well-established risk factors include air pollution, smoking, poor diet, hypertension, and physical inactivity [3, 4], which explain most but not all of the attributable cardiovascular disease burden. It is thought that precipitating factors such as mental stress, physical exertion, certain drugs and infections may 'trigger' susceptible individuals to experience AMI or stroke [5]. Gaining insight into the role of these factors is crucial for improved risk prediction and prevention.

Given the prevalence of infections, strong biological plausibility, and apparent reduction in acute cardiovascular outcomes observed in influenza vaccine randomised controlled trials [6–8], the contribution to the burden of cardiovascular disease may be substantial. To date, reliable data are largely obtained from epidemiological studies focusing on individuals with influenza infection [9]; however, other small-scale studies have described the effects of various bacterial and viral infections on specific cardiovascular endpoints [10–12]. Infections involving the lower and upper respiratory tract are most frequently implicated as triggers for AMI and stroke; attributable respiratory viruses include influenza, respiratory syncytial virus (RSV) [10, 11], cytomegalovirus (CMV) [12], and more recently, severe acute respiratory syndrome coronavirus 2 (SARS-CoV-2) [13]. Respiratory viruses can infiltrate the lungs through airway epithelial cells, directly stimulating the production of proinflammatory cytokines. The hypothesis is that this triggers systemic inflammation, inducing endothelium damage, atherosclerosis and plaque rupture [14], increasing the risk of AMI and stroke [15]. The duration of the risk period following respiratory infection is typically brief, lasting from several days to a few months, and gradually diminishes with reduced viral load [16, 17].

Given the evidence of a protective relationship between influenza vaccine and cardiovascular disease risk [18, 19], high-quality observational studies of other viral precipitants could pave the way for future trials beyond influenza. However, the comparative pathogen-specific effects of respiratory viruses on the risk of AMI and stroke remain unclear. Modern diagnostic

tests, such as increasingly available multiplex respiratory panels, enable the distinction between potentially attributable viruses. Previous systematic reviews and meta-analyses have focused solely on a single viral trigger like influenza or SARS-CoV-2 [20, 21], or relied on broad definitions of infection (e.g. symptomatic respiratory tract infection or influenza-like illness) [22], making interpretation challenging as data were not pooled across studies detecting different attributable viruses [23]. In this systematic review, we aim to examine the evidence from studies of respiratory viruses identified by laboratory-confirmatory testing, compare the associations between viral respiratory triggers, and estimate the pooled pathogen-specific effects on acute myocardial infarction and stroke events.

## Objectives

Our objective is to systematically review the associations between acute myocardial infarction/ stroke and laboratory-confirmed respiratory viral infection. This will be addressed through a comprehensive search of analytical epidemiological evidence in humans of any age or geographical region, specifically published studies that capture laboratory-confirmed respiratory viral infections and their effects on acute myocardial infarction or stroke outcomes. A secondary objective is to identify gaps in the literature, assess the certainty of the evidence, and recommend areas for future research.

## Methods

The systematic review has been registered with the International Prospective Register of Systematic Reviews (PROSPERO): CRD42024494997. If there are protocol amendments, important changes will be explained (including the date and reason for the amendment) in the completed published review, in addition to tracking in PROSPERO.

### Eligibility criteria

The eligibility criteria based on the Population, Intervention/Exposure, Comparison, Outcome, Study design, and Timeframe (PICOST) approach, with "intervention" replaced by "exposure", are as follows:

### Population

Studies involving human participants of any age in any country or region. All healthcare or community settings (e.g. hospital admission, emergency department, ambulance attendances, and primary care consultations) will be considered.

**Exposure.** Laboratory-confirmed respiratory viral infection. We will consider viruses that primarily cause illnesses affecting the nose, throat, mouth, and breathing passages, such as common cold or flu-like symptoms. Respiratory viruses include (but are not limited to): influenza, parainfluenza, RSV, human metapneumovirus, rhinovirus, adenovirus, and SARS-CoV-2. Bacteria (including mollicutes) or fungi are excluded. Studies will only be included in which the specific causative microorganism(s) are reported, as detected by confirmatory laboratory methods, including polymerase chain reaction (PCR) and serological tests, but exclude point-of-care testing such as rapid antigen tests. Suspected infection, influenza-like illness, or unspecified respiratory tract infections in the absence of laboratory-confirmed results, are excluded. We will exclude viruses that do not primarily result in respiratory phenotypes, such as human immunodeficiency virus (HIV), hepatitis viruses, dengue virus, Ebola virus, and other viral haemorrhagic fevers. Latent or persistent viral exposure, such as human herpesviruses detected solely by serum IgG antibodies, will be excluded.

**Comparator.** Eligible studies must include a comparator group that is unexposed, that is, without respiratory infection, negative testing individuals, or non-active/latent infection. This includes individuals with negative laboratory test result(s), an infection-free group in cohort and case-control studies, or unexposed person-time in case-only studies. In randomised controlled trials (RCTs) testing specific anti-infective interventions, such as influenza vaccine studies, data will be treated observationally, that is, the comparator group will be considered regardless of treatment allocation. Before and after (pre–post) COVID-19 pandemic comparators are excluded to ensure all groups have similar chance of exposure to the circulating respiratory virus.

**Outcome.** Fatal and non-fatal incidence of AMI or stroke (ischaemic or haemorrhagic), either first or recurrent events. Studies will be included if both or one of these conditions is measured, that is, either AMI or stroke. Diagnosis may be clinical, routinely coded, or self-reported. Patients with composite acute cardiovascular or cerebrovascular outcomes, but not exclusively AMI or stroke, will be excluded. We will exclude transient ischaemic attacks and other neurological or thrombotic conditions in which AMI or stroke is not specifically mentioned, such as unstable angina, cardiac arrest, congestive heart failure or cardiomyopathy, peripheral arterial disease, or venous thromboembolism. Cardiovascular or cerebrovascular inflammatory conditions of infectious origin, such as myocarditis, encephalitis, meningitis, or endocarditis, will be excluded. RCTs capturing adverse events following vaccination and surgical or pharmacological complications will be excluded if AMI or stroke is not the primary outcome. Studies assessing only intermediate outcomes such as inflammatory biomarkers are also excluded.

**Study design.** Primary research papers describing RCTs, cohort and case-control studies, self-controlled case series, and case crossover designs. RCTs will be included where there are arm-specific data for the effect of infection on the risk of incident AMI and/or stroke. Cross-sectional study designs, uncontrolled before-after studies including interrupted time series, ecological studies, case reports, or case series of individual patients, and reviews will be excluded. Studies must report the effect estimates (risk ratio, odds ratio, rate ratio, incidence rate ratio, or hazard ratio) or provide data to calculate an estimate of the effect on the risk of acute myocardial infarction and/or stroke.

**Timeframe.** While there are no time restrictions on the risk window or follow-up period used, we will prioritise studies using short-term risk periods (up to 90 days) of laboratory confirmation i.e. primary outcome occurring within the date of specimen collection. Studies with longer risk periods will be assessed for eligibility considering the risk of bias for inclusion. This time window focuses on the cardiovascular triggering effect following viral infection. We will consider contacting corresponding authors to confirm eligibility or to obtain further information where needed.

No date, country, geographical, language, publication status or minimum sample size restrictions apply. Studies in which the methods are insufficiently described or without extractable results will be excluded. Refinement of the eligibility criteria is expected to be iterative and depends on the search results identified during screening. We will consider contacting authors for clarification or if further information is needed to confirm eligibility. This systematic review will be reported in accordance with the PRISMA guidelines and Meta-analysis of Observational Studies in Epidemiology (MOOSE) statement (if applicable) [24, 25]. Additionally, the current protocol adheres to the Preferred Reporting Items for Systematic Review Protocols (PRISMA-P) 2015 statement (see S1 File for checklist) [26].

## Information sources

We will search the following electronic bibliographic databases from their inception until the end of March 2024: MEDLINE (Ovid), Embase (Ovid), PubMed (NLM), Cochrane Central

Register of Controlled Trials (Wiley), and Web of Science Core Collection (Clarivate Analytics). The database selection and search strategy were developed in close consultation with an experienced medical/healthcare librarian engaged in the early planning phase. In addition, we will check the reference lists of the included studies (backward citations) and articles that cited the included studies (forward citations) to identify any further relevant papers. Where necessary, investigators will be contacted to request further information about conference abstracts, trial registrations, or unpublished work. A preliminary search of previous systematic reviews on the topic to identify existing or related reviews registered on PROSPERO and via the Cochrane Library has been conducted to avoid duplicating work. Articles cited in previous systematic reviews will be checked to identify any other potentially relevant studies.

## Search strategy

The search strategy combines thesaurus terms or Medical Subject Headings (MeSH), where available, and free-text synonyms for search concepts based on the PICOST framework (S2 File). A search strategy has been designed based on the Population, Exposure, Outcome and Study design elements using Boolean operators (AND, OR) to combine variations of the following search concepts: "Humans", excluding animal studies; "Laboratory confirmatory tests", including PCR assays, serology, viral culture and immunofluorescence; "respiratory viruses" including influenza, coronaviruses including SARS-CoV-2, picornaviruses (including enteroviruses), human metapneumovirus, RSV, paramyxoviruses (including parainfluenza viruses), adenoviruses, herpes family viruses (including herpes simplex virus 1 and 2, varicella zoster virus, Epstein-Barr virus and cytomegalovirus) and human bocavirus; "acute myocardial infarction or stroke" including ischaemic stroke, haemorrhagic stroke; and from analytical epidemiological study designs. Where thesaurus terms or MeSH exist, for example, virus families covering relevant species or subtypes, these have been exploded to include all narrower terms in the tree (if appropriate). The search includes methodological search filters adapted from the BMJ Best Practice strategies, designed to reliably retrieve specific study designs [27]. The strategy has been developed initially for MEDLINE via the Ovid interface, then subsequently translated into appropriate search terms for the other databases. The Polyglot Search Translator tool was used to assist with translation across from MEDLINE into the Cochrane Library [28].

## Study records

**Data management.**  Citations retrieved will be uploaded to Covidence, an online software system used to manage systematic reviews and promotes collaboration among authors [29]. Duplicates will be removed using the in-built duplicate feature, and further deduplication by manual review in Covidence. Upon completion of title and abstract screening, full-text articles will be uploaded to Covidence for full-text review. The reference management software End-Note X21 (Clarivate Analytics) will be used to manage records outside of Covidence throughout the review [30].

**Selection process.**  The title/abstract screening process will initially be piloted with at least 50 articles to ensure consistent implementation of the eligibility criteria between reviewers. Following deduplication, the search results will be screened for eligibility by two independent reviewers in two stages: title and abstract screening, followed by a review of the full text. Discrepancies in either of the screening stages will be resolved by consensus and/or arbitration by a third reviewer. Inter-rater reliability will be calculated using the Cohen κ value for both stages of screening. Studies determined to be ineligible will be excluded from the review and reasons for exclusion at the full-text stage reported in accordance with Preferred Reporting

Items for Systematic reviews and Meta-analyses (PRISMA) [24]. A PRISMA-compliant flow chart will be used to display the selection of articles with reasons for exclusion.

**Data collection process.**   Data will be extracted using a standardised template based on the PICOST framework using Covidence software. A pre-defined data collection form in Covidence will be piloted on at least 5 of the included studies, and changes to the form will be made if required. We will collect data on study populations, settings, exposure and outcome definitions, outcome measures, effect sizes, confidence intervals, and results of significance testing. The authors of the studies will be contacted if missing data are identified during the data extraction phase.

We will assess the need for double extraction of results if there is sufficient homogeneity for meta-analysis. If meta-analysis is required, we will use double data extraction (of critical data items required for the interpretation of results) performed independently by two people to minimise errors, and discrepancies will be resolved by discussion. Otherwise, data extraction will be performed by one reviewer and checked for accuracy by a second reviewer.

## Data items

Data items on the following five domains will be extracted:

1. Study characteristics: author(s), publication, year, country, study design, funding source(s)

2. Population: characteristics of the study population (e.g., age, sex, country and setting, inclusion and exclusion criteria), sample size, study period, follow-up time (if applicable)

3. Exposure: viral respiratory pathogen(s), definition of exposure(s), risk window used, type of laboratory test used, number of exposed subjects

4. Comparators: definition of unexposed individuals, number of unexposed subjects, confounders adjusted for

5. Outcomes: acute myocardial infarction and/or stroke, outcome definition and method of diagnosis, disease subtype (if applicable), number of subjects with outcome, effective size, 95% confidence interval, p-value

For studies meeting the inclusion criteria, we will additionally assess mortality outcomes and data among the specific cardiovascular endpoints: ST-elevation myocardial infarction (STEMI) or non-STEMI, and ischaemic or haemorrhagic stroke, if available.

## Risk of bias in individual studies

Study-level methodological quality assessment will examine the strength of the evidence from individual studies regarding the presence and nature of potential triggering effect(s) of respiratory virus exposure on acute cardiovascular outcomes. We will use the Cochrane Collaboration tools for randomised trials and non-randomised studies, the Cochrane Revised Risk of Bias tool for randomised trials (RoB 2) and the Cochrane Risk of Bias in Non-randomised studies–of Exposures (ROBINS-E) for observational studies to judge overall risk of bias for each individual study based on study methodology [31]. For RCTs, domains will include bias arising from 1) the randomisation process, 2) deviations from intended interventions, 3) missing outcome data, 4) measurement of the outcome and 5) selection of the reported result [32]. For non-randomised studies, the ROBINS-E tool for assessing epidemiological studies of exposure-outcome effects assesses seven domains for bias due to 1) confounding, 2) measurement of the exposure, 3) selection of participants into the study or analysis, 4) post-exposure

interventions, 5) missing data, 6) measurement of the outcome and 7) selection of the reported result [33].

The risk of bias assessment of the studies will be performed independently by two reviewers, and discrepancies will be resolved through discussion to achieve consensus. However, failing agreement, a third reviewer arbitrates. The overall risk of bias judgement will be categorised as 'low risk of bias', 'some concerns', or 'high risk of bias'. The results will be synthesised into a narrative summary and incorporated into the primary analysis, stratified by outcome and sub-groups, if appropriate. A summary table of domain-level judgments will be produced to show how each study fares.

## Data synthesis and meta-bias(es)

All syntheses will be categorised according to their primary outcome (acute myocardial infarction or stroke). Study characteristics and measured outcomes will be compiled into summary tables grouped by outcomes and specific viral pathogens. If the studies are sufficiently homogenous by outcome definition and population, we will combine the data statistically in a meta-analysis. A prerequisite is that the outcome definitions and relative effect measures need to match, and that at least three or more studies report the same outcome to trigger a meta-analysis.

For studies of the same primary dichotomous outcomes (acute myocardial infarction or stroke), a forest plot visualisation including confidence intervals, Cochrane's Q test for statistical heterogeneity, and $I^2$ statistic will be used to assess between-study heterogeneity [34]. Where appropriate, meta-analysis will be conducted to calculate pooled effect sizes in a fixed-effect or random-effects model.

We will investigate sources of heterogeneity for each primary outcome by:

a. Sub-group analysis, and

b. Meta-regression, that is, meta-analyses for each subgroup (study design, outcome definition, studies with good/poor control for confounding, effects in older/younger individuals, etc).

Factors that may contribute to heterogeneity between studies evaluating the same outcome include: methodological diversity (study design, data source, or setting), study population characteristics (age strata, sex, country, socioeconomic status, and comorbidities), exposure definition (specific virus, laboratory test, or risk window), interventions (vaccinated or unvaccinated), outcome definition (disease, subtype), and diagnosis (clinical diagnosis, administrative data, or self-report).

Where meta-analysis is not appropriate, only a qualitative synthesis is presented. Publication bias will be assessed using funnel plots. If sufficient data are available in the included studies, we will synthesise the answers to our research questions by subgroup. All analyses will be performed in R Statistical Software (v4.3.2 or later) [35].

## Confidence in cumulative evidence

The Grading of Recommendations, Assessment, Development and Evaluation (GRADE) approach will be used to assess the certainty of the evidence for each respiratory virus on each individual outcome, by patient population, across the five domains (risk of bias, inconsistency, indirectness, imprecision and publication bias) for population-level outcomes [36]. The strength of the evidence will be categorised as 'low', 'moderate' or 'high', with observational studies starting as low-quality evidence, but upgraded to moderate or high quality in the presence of other factors that increase confidence in the result e.g. strong association, evidence of

dose response gradient. Owing to the life-threatening, critical nature of our primary outcomes, we consider consistent nonzero relative effects (for example, RR>1) and precise estimates to be important. We will report the GRADE assessment in our results stratified by type of exposure with a row for each important outcome (myocardial infarction, stroke, or specific subtype), and the strength of the evidence will be presented in a 'Summary of findings' table. This assessment will enable formal judgement of the overall quality of the evidence included in the review, summarising gaps in the literature, and areas for future research.

## Ethics and dissemination

This review is ongoing. As this is a systematic review, ethical approval is not required. Upon completion, the results will be submitted to a relevant journal for peer-reviewed publication, presented at scientific conferences in this field. A lay and short summary will be displayed in appropriate reports.

## Limitations

This systematic review may have some limitations. In general, observational study data are affected by inherent design-specific biases, such as confounding and lack of temporality; hence, robust risk of bias assessments were factored into our synthesis plan. As we are including a broader range of viral exposures relative to other systematic reviews there may be some exposure-outcome relationships described by only a few included studies or small samples within those studies. This can potentially be mitigated by our comprehensive database coverage and highly inclusive search strategy, but data may be limited for less commonly tested respiratory viral pathogens. Finally, it is expected that there will be high levels of heterogeneity between different stroke subtypes, specifically haemorrhagic and ischaemic stroke.

## Conclusion

Implementing cost-effective policies to avoid premature deaths, disability and unsustainable economic costs attributable to acute myocardial infarction and stroke are a global health priority [37]. There is currently indirect epidemiological evidence regarding the immediate triggering effects of influenza and other respiratory viruses, but the pooled pathogen-specific risk is unknown. Given the co-circulation of respiratory viruses and their increasing ability to prevent and mitigate symptomatic infections, it is important to quantify the relative effects of causative agents, particularly respiratory viruses, in order to evaluate the risk of cardiovascular outcomes. The findings of this systematic review offer value to clinicians as it will provide comprehensive evidence regarding common respiratory virus exposure and risk among high-risk patients. The findings will help to identify candidate pathogens for potential vaccine trials or other preventive therapies in targeted populations. Furthermore, where attributable pathogens have established vaccines, this could be used for potential vaccine probe studies to inform vaccine-preventable cardiovascular burden [38].

## Supporting information

**S1 File. PRISMA-P checklist.** Preferred Reporting Items for Systematic review and Meta-analysis Protocols (PRISMA-P) checklist.
(DOCX)

**S2 File. Search strategy.** Search strategy for Ovid MEDLINE, Ovid Embase, PubMed, Cochrane Central Register of Controlled Trials (CENTRAL), and Web of Science.
(DOCX)

## Acknowledgments

The authors would like to thank Poh Chua (Librarian, Royal Children's Hospital Melbourne), whose advice and guidance were of integral importance in planning and designing the search process from the start of the protocol, and Lai-Yang Lee for providing technical expertise in interpreting latent viral diagnostics (Medical Microbiologist, Royal Children's Hospital Melbourne).

## Author Contributions

**Conceptualization:** Tu Quan Nguyen, Diana Vlasenko, Aishwarya N. Shetty, Jim P. Buttery.

**Funding acquisition:** Jim P. Buttery.

**Investigation:** Tu Quan Nguyen, Diana Vlasenko, Aishwarya N. Shetty.

**Methodology:** Tu Quan Nguyen, Diana Vlasenko, Aishwarya N. Shetty, Hazel J. Clothier.

**Supervision:** Christopher M. Reid, Hazel J. Clothier, Jim P. Buttery.

**Writing – original draft:** Tu Quan Nguyen.

**Writing – review & editing:** Tu Quan Nguyen, Diana Vlasenko, Aishwarya N. Shetty, Christopher M. Reid, Hazel J. Clothier, Jim P. Buttery.

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
