## [Decision Letter · Decision Letter 0]

29 May 2024

PONE-D-24-14603Laboratory-confirmed respiratory viral infection triggers for acute myocardial infarction and stroke: systematic review protocolPLOS ONE

Dear Dr. Nguyen,

Thank you for submitting your manuscript to PLOS ONE. After careful consideration, we feel that it has merit but does not fully meet PLOS ONE’s publication criteria as it currently stands. Therefore, we invite you to submit a revised version of the manuscript that addresses the points raised during the review process.

We look forward to receiving your revised manuscript.

Kind regards,

Zhehao Dai

Academic Editor

PLOS ONE

Additional Editor Comments:

The manuscript is a well written protocol for systematic review. Please revise the manuscript as suggested by the reviewers, before this manuscirpt would be considered for publication.

Reviewers' comments:

Reviewer's Responses to Questions

**Comments to the Author**

1. Does the manuscript provide a valid rationale for the proposed study, with clearly identified and justified research questions?

Reviewer #1: Yes

Reviewer #2: Yes

2. Is the protocol technically sound and planned in a manner that will lead to a meaningful outcome and allow testing the stated hypotheses?

Reviewer #1: Yes

Reviewer #2: Yes

3. Is the methodology feasible and described in sufficient detail to allow the work to be replicable?

Reviewer #1: Yes

Reviewer #2: Yes

4. Have the authors described where all data underlying the findings will be made available when the study is complete?

Reviewer #1: Yes

Reviewer #2: No

5. Is the manuscript presented in an intelligible fashion and written in standard English?

Reviewer #1: Yes

Reviewer #2: Yes

6. Review Comments to the Author

You may also provide optional suggestions and comments to authors that they might find helpful in planning their study.

Reviewer #1: The authors present a systematic review protocol to systematically review the associations between acute myocardial

infarction/stroke and laboratory-confirmed respiratory viral infection. This protocol was well constructed and follows the PRISMA-P recommendations. Minor corrections must be made before being published.

In line 107, authors must enter the protocol number registered with PROSPERO. In addition, they must correct repeated text: "for the effect of infection for the effect of infection", in line 155.

Finally, before the conclusion, the authors must include a paragraph with potential limitations of the future systematic review, such as small number of studies, small samples in existing ones and expected biases in observational studies, explaining how they intend to deal with the limitations.

Reviewer #2: Thank you for the opportunity to review this protocol! My expertise is review methods generally and systematic searching specifically so that's what I will be focusing on in my peer review.

Abstract

Your abstract overall is very clear. Though you do describe a lot of publication-based eligibility criteria, the clinical criteria is kind of lacking (e.g. which infections, what tests, population groups, MI, stroke) which I think would bring added clarity.

Methods

Love that you describe the changes from your registered protocol. I'm surprised though that these aren't addressed in this protocol rather than the published review. Unless I missed it? Or you mean *if* there are any amendments? Clarity here might be helpful.

Your searches look excellent, thank you for providing all of them in full. I notice though that they aren't showing results line by line - this would be helpful to see as I wonder whether your study design filters are needed, since you're interested in many different study designs and adding these filters potentially adds bias to your search. If the difference in results is not great they may not be warranted.

You mention a librarian a few times in this protocol, and I see them acknowledged below. Did they write the search? If so, why are they not also an author on this project?

You could probably cut down the Search strategy section of the protocol, as it goes into an unnecessary level of detail.

Will the search for this review be peer-reviewed? Please describe.

Data

Given PLoS' data policy, what are the team's plans for making data available such as full list of full-text exclusions, extraction data, and critical appraisal/quality checklists available?

Final thoughts: This is a very well written protocol. The searches are robust and there is good database coverage. Some minor tweaks needed.

7. PLOS authors have the option to publish the peer review history of their article (what does this mean?). If published, this will include your full peer review and any attached files.

Reviewer #1: **Yes: **Ricardo Ney Cobucci

Reviewer #2: No

---

## [Author Response · Author response to Decision Letter 0]

13 Jun 2024

Uploaded as a separate file 'Response to Reviewers'

---

## [Decision Letter · Decision Letter 1]

26 Jun 2024

Laboratory-confirmed respiratory viral infection triggers for acute myocardial infarction and stroke: systematic review protocol

PONE-D-24-14603R1

Dear Dr. Nguyen,

We’re pleased to inform you that your manuscript has been judged scientifically suitable for publication and will be formally accepted for publication once it meets all outstanding technical requirements.

Kind regards,

Zhehao Dai

Academic Editor

PLOS ONE

Reviewers' comments:

Reviewer's Responses to Questions

**Comments to the Author**

1. Does the manuscript provide a valid rationale for the proposed study, with clearly identified and justified research questions?

Reviewer #1: Yes

Reviewer #2: Yes

2. Is the protocol technically sound and planned in a manner that will lead to a meaningful outcome and allow testing the stated hypotheses?

Reviewer #1: Yes

Reviewer #2: Yes

3. Is the methodology feasible and described in sufficient detail to allow the work to be replicable?

Reviewer #1: Yes

Reviewer #2: Yes

4. Have the authors described where all data underlying the findings will be made available when the study is complete?

Reviewer #1: Yes

Reviewer #2: Yes

5. Is the manuscript presented in an intelligible fashion and written in standard English?

Reviewer #1: Yes

Reviewer #2: Yes

6. Review Comments to the Author

You may also provide optional suggestions and comments to authors that they might find helpful in planning their study.

Reviewer #1: The authors met the reviewers' recommendations and the manuscript is ready to be published. Congratulations!

Reviewer #2: Thank you for your revisions! Unfortunately I was unable to locate the Response to Reviewers document on the platform but your revised manuscript has addressed the majority of my comments.

7. PLOS authors have the option to publish the peer review history of their article (what does this mean?). If published, this will include your full peer review and any attached files.

Reviewer #1: **Yes: **Ricardo Ney Cobucci

Reviewer #2: No

---

## [Editor Report · Acceptance letter]

28 Jun 2024

PONE-D-24-14603R1 

PLOS ONE

Dear Dr. Nguyen, 

I'm pleased to inform you that your manuscript has been deemed suitable for publication in PLOS ONE. Congratulations! Your manuscript is now being handed over to our production team.

Kind regards, 

on behalf of

Dr. Zhehao Dai 

Academic Editor

PLOS ONE